# A Pilot Study of Family-Integrated Care (FICare) in Critically Ill Preterm and Term Infants in the NICU: FICare Plus

**DOI:** 10.3390/children10081337

**Published:** 2023-08-02

**Authors:** Najmus Sehr Ansari, Linda S. Franck, Christopher Tomlinson, Anna Colucci, Karel O’Brien

**Affiliations:** 1Department of Pediatrics, University of Toronto, Toronto, ON M5S 1A1, Canada; christopher.tomlinson@sickkids.ca (C.T.); karel.obrien@sinaihealth.ca (K.O.); 2Division of Neonatology, The Hospital for Sick Children, Toronto, ON M5G 1X8, Canada; 3School of Nursing, University of California, San Francisco, CA 94143, USA; linda.franck@ucsf.edu; 4Department of Pediatrics, Mount Sinai Hospital, Toronto, ON M5G 1X5, Canada; anna.colucci@sinaihealth.ca

**Keywords:** family-integrated care, critically ill neonates, safety, feasibility

## Abstract

Family-integrated care (FICare) is associated with improved developmental outcomes and decreased parental mental health risks in stable preterm infants. However, less is known about its application in critically ill infants who are at greater risk for adverse outcomes. The objective of this study was to assess the safety and feasibility of implementation of an augmented FICare program, FICare Plus, in critically ill infants in the first few weeks of life. Resources were specifically developed for staff and parents to support earlier parental engagement in infant care. Infant health outcomes and standardized measures of parental stress, anxiety and parenting self-efficacy were also collected using standardized questionnaires: State -Trait Anxiety Inventory (STAI), Parental Stressor Scale: NICU (PSS: NICU), Perceived Parenting Self-Efficacy Tool and Family Centered Care Survey. The *t*-test or Wilcoxon rank-sum test were used to compare continuous variables, while the Chi-square or Fisher exact test were used for categorical variables, respectively. In this prospective cohort study, 41 critically ill infants were enrolled: 17 in standard care (SC) and 24 in the FICare Plus group. The tools and procedures developed for FICare Plus successfully supported greater engagement in the care of their infants with no increase in adverse events and no increase in parental stress. Parents in the FICare Plus cohort felt confident to participate in their infant’s care. The staff also found this model of care acceptable and well adopted. Preliminary measures of infant efficacy were similar in both groups. Total anxiety scores were high among all parents at enrollment (87 (67–94) vs. 70.5 (66–86); *p*-value 0.22). However, the scores prior to discharge were lower in FICare Plus group (78 (71–90) vs. 63 (52–74.5); *p*-value 0.02). This pilot study showed that it is feasible and safe to implement family-integrated care in critically ill infants.

## 1. Introduction

Family-integrated care (FICare) involves engaging parents in their infant’s care and supporting them to be active participants in all aspects of infant care planning and caregiving, in partnership with healthcare professionals [1]. FICare is a care delivery model that addresses the structure and processes of care delivery in neonatal intensive care units (NICUs). It provides a framework for parent-partnered care delivery that can include multiple specific parent-partnered interventions focused on different aspects of care, for example, feeding and infant development [2]. Studies conducted in different countries and health systems have shown improved neonatal outcomes and enhanced psychological and emotional well-being in parents [1,3,4]. Clinical trials have shown improved weight gain, shorter hospitalization, lower rates of infection and higher rate of breastfeeding at discharge in infants who received family-integrated care [5,6]. Furthermore, there is a growing body of evidence which suggests that FICare improves long-term neurodevelopmental outcomes in infants [7,8].

The original studies of FICare enrolled preterm infants born less than or equal to 35 weeks gestation who were stable on non-invasive ventilation at the time of enrollment although many were born very preterm [1]. Investigators have also reported the outcomes of implementing FICare in level 2 NICUs [9]. However, limited evidence is available about the feasibility of implementing FICare with families of infants who are critically ill at the time of enrollment. Only one study has investigated FICare in critically ill preterm neonates in the United States [10]. Although there were no group differences in outcomes commonly measured in prior studies of stable preterm infants, there were no adverse effects, and several FICare components (rounds, peer parent mentors and group parent classes) were associated with improved outcomes compared with standard family-centered care [10]. Another study implemented a modified FICare program NICU-wide in a community level hospital, including some term and preterm infants. In this study, nurses reported positive views about the program, and parents had reduced stress and improved readiness for discharge after implementation of the program [11].

The technologically advanced environment of neonatal intensive care units can often be very daunting and stressful for parents [12,13]. This stress is further heightened by their infant’s fragility and their limited ability to participate as a caregiver, which may adversely impact their confidence to acquire normal parenting skills [14,15,16]. Although critically ill infants may have greater gains from FICare, there was a concern, especially from our family advisors, that implementing FICare in this population might cause unintended distress to parents. Although it may have seemed like a natural extension to provide FICare to all families, concerns were brought forward by both staff and parent advisors as to whether this was the correct approach for parents of critically ill infants. Parents brought forward concerns that the expectations around early engagement in their infant’s care might be overwhelming for them. Staff were concerned about increasing parental anxiety and the possibility of increased medical risk to the infants from parent engagement while still unstable. There were no published data in this population at the time of study conception. Facilitating infant–parent interaction creates a consistent care environment which is beneficial for both infants and their parents [5]. Moreover, parental confidence and expertise in taking care of their infants, particularly if they have complex needs, may lead to better long-term neurodevelopmental outcomes in their infants [8]. Given the potential benefits for critically ill infants and their families, we set out to adapt FICare to accommodate the needs of this vulnerable population. Therefore, we hypothesized that, despite these concerns, parents of critically ill infants might in fact be the group who, if well supported, may benefit more from FICare. This study was conducted with the objective to assess if it is safe and feasible to support greater parent engagement in taking care of their critically ill infants, without increased parental stress and anxiety or increasing risk of harm to infants.

## 2. Materials and Methods

### 2.1. Study Design

This was a prospective cohort study, pre/post-implementation, of an augmented FICare program for critically ill infants of all gestational ages, termed as FICare Plus. It was conducted in two tertiary care NICUs, Mount Sinai Hospital (Site A) and SickKids Hospital (Site B) in Toronto, Canada. Site A is a tertiary care perinatal center with more than 1100 infant admissions per year to the NICU. Of these infants, 13% are extremely low-birth-weight (ELBW) (<1000 g) infants. Site B is a quaternary care referral center with an average of 800 admissions/year to the NICU. Approximately one-third of these infants require surgery during the neonatal period. In addition to serving different infant populations, these two NICUs have different levels of experience with FICare. Site A has over 10 years of experience, whereas Site B has not implemented FICare as their standard of care.

To assess the safety and feasibility of implementation of FICare Plus in critically ill infants, data were collected in a sequential manner, pre- and post-implementation. This design was considered most feasible for this study as families could not be randomized to different models of care within the same unit. The study protocol, questionnaires and the educational toolkit developed for this study were approved by the research ethics board at both hospitals.

### 2.2. Participants

Infants eligible for this study included: ELBW infants on invasive positive-pressure ventilation for >48 h after birth, infants with surgical necrotizing enterocolitis or bowel perforation, infants with tracheoesophageal fistula or esophageal atresia. Families were approached while the infant was <2 weeks of age. For both cohorts, families were asked to commit to being present at the hospital for 4 h per day, for a minimum of 5 days/week. For parents of multiples, families were only asked to commit to the same time commitment even though their time might be divided between their infants. Infants were excluded if they were receiving palliative care, were deemed unlikely to survive either due to critical illness or other life-threatening congenital anomaly or were born to parents with inability to participate either due to health, social or language issues. Parents in the FICare Plus group were encouraged to attend and participate in daily morning rounds. They were encouraged to engage in their infant’s care under nursing supervision. Written informed consent was obtained from all enrolled participants. Parents in both groups were asked to complete questionnaires at specific time points including their demographic information.

### 2.3. Procedures

Pre-implementation: In the development phase of this study, teams were formed at both sites which included parents, bedside nurse/s, a respiratory therapist, a social worker, physicians, and an educator to plan and execute this project at their respective sites. The nursing staff, veteran parents (through Canadian Premature Babies Foundation) and current parents were surveyed at both hospitals to determine how parents could safely and meaningfully be involved in the care of their critically ill babies. The information gathered from these surveys was reviewed in a joint meeting of both site teams.

FICare Plus educational toolkit for staff and parents: The teams identified critical opportunities for creating a connection with families at both sites and went on to develop educational resources for parents and staff (will be uploaded to the FICare website (https://familyintegratedcare.com)). From the parent survey, it was identified that parents might have difficulty attending group classes in the immediate days following admission of their infant to the NICU. In response, a toolkit was developed for both sites by consensus, which included a parent booklet and online educational resources, including videos for parents to access in their own time. It was also recognized that staff might need additional education so that they could better support parents, particularly when their infant was just admitted to the hospital. In response, an on-line staff educational booster toolkit was developed, comprising short videos providing information about the importance of parent engagement in infant care, parent reflections on their experience in the NICU, and coaching staff about how best to support parents (see https://familyinteratedcare.com). A detailed inventory of potential parent and nurse responsibilities for the FICare Plus group was created and reviewed at both sites, recognizing that there could be some site-specific accommodations depending on the needs of the infants. While the resources for implementation were being developed, participants were enrolled in the standard care (SC) cohort at both sites.

The SC cohort received the usual care at both sites. FICare Plus training was provided to nursing staff after the accrual of the SC cohort so as not to influence the SC provided to this group.

Implementation: Once recruitment to the SC group was completed, FICare Plus booster training was provided to >90% of nursing staff at both sites. The NICU policies and procedures at both sites were reviewed and revised to enable greater parent participation from admission. Between 1 January 2021 and 31 July 2021, parents were enrolled in the FICare Plus group. Parents were oriented to the tools provided for their self-education by a specially trained FICare Plus nurse educator.

Parents were supported by their bedside nurse to participate in their infant’s care and were taught ways to interact with their infants which may optimize their development. They were also provided with information about coping strategies and stress-reducing activities, in addition to the routine support provided by the social worker. Enrollment was maintained to ensure equal representation of infants in the observational and the implementation cohorts.

The research coordinators at both sites ensured that all enrolled families were able to access the educational materials provided, were being offered to participate in their infant’s care and were supported to participate in rounds. Since families in both cohorts were recruited during the COVID-19 pandemic, research coordinators had limited access to both NICUs and therefore had limited ability to approach families for enrollment. A single coordinator approached all families for all the concurrent studies in the unit and then referred those who were interested in hearing more about a specific study to the respective coordinator. Our FICare Plus study coordinators were also unable to have face-to-face contact with families. They provided support and encouragement to families to complete the data collection through phone calls. In addition, peer support, which has been a cornerstone of FICare implementation in our other studies, was not provided over this time period. The parent ongoing education sessions, which were usually standard at Site A were provided inconsistently and virtually rather than in person.

### 2.4. Data Collection

The primary aim of this study was to assess if it was feasible to support greater parent engagement in taking care of critically ill infants given the tools and procedures developed for FICare. The measures of this outcome included (1) the percentage of approached parents who enrolled in the pilot study, (2) parent access to and evaluation of on-line education modules and educational materials, (3) the parents’ perceptions of their engagement in their infant’s care as indicated on their parent surveys. Secondary outcomes included the safety of the intervention as measured by (1) the number of reported adverse events related to increased parental involvement in their infant’s care, (2) the acceptability of FICare Plus as reported by the multidisciplinary team on their surveys (3) whether there was any difference between the staff responses at both sites.

Preliminary measures of efficacy were collected pre- and post-implementation and included both infant and parent measures. The infant measures were the type of feeding at discharge (tube feeding, exclusive breast milk feeding, exclusive formula feeding or a combination of formula/breast milk), weight at discharge and discharge destination. Parent measures included measures of parental stress and anxiety, parent perceived competence in parenting and family-centered care. Stress is defined as a physiological response to a stimulus which is accompanied by a simultaneous emotional reaction, whereas anxiety is defined as “the anticipation of a future threat” by The Diagnostic and Statistical Manual of Mental Disorders, Fifth Edition (DSM-5) [17]. Parent anxiety was measured by the State-Trait Anxiety Inventory (STAI) [18] at enrollment, day 21, prior to discharge and one week post discharge. The STAI is a 40-item questionnaire with scores ranging from 1 (lowest anxiety) to 4 (highest anxiety) for each item with cumulative scores ranging from 40–160. Parental stress was assessed using the Parental Stressor Scale: NICU (PSS: NICU) [19] which is a validated scale to assess parental stress during NICU hospitalization of their infant; the average overall scores range from 1–5 in order of severity. Parental perceived parenting self-efficacy was measured using Perceived Parenting Self-Efficacy Tool [20]. The score range in this validated tool is 20–80, with higher scores indicating higher perceived self-efficacy. Parents’ NICU experience was assessed using the Family-Centered Care Survey [21]. Their response was rated on a 7-point scale with higher values indicating a more positive experience. Linked parent and infant demographic data were extracted from the Canadian Neonatal Network database.

### 2.5. Statistical Analysis

The data analysis was based on an intention-to-treat basis. The measures of both the primary and secondary outcomes are presented with calculated descriptive statistics. Comparisons by intervention group were performed using the *t*-test or Wilcoxon Rank-Sum test for continuous variables, and Chi-square or Fisher exact test for categorical variables. For the group comparisons, *p*-value < 0.05 was considered statistically significant. All analyses were performed using SAS version 9.4 (Cary, NC, USA).

## 3. Results

Between 1 January 2020, and 31 December 2020, a total of 115 infants were assessed at both hospitals for eligibility in the SC cohort (Figure 1). Of the 36 families who were approached, 17 parent–infant dyads (47.2%) were enrolled in the SC cohort, between the two sites. Between 1 January 2021 and 31 July 2021, a total of 245 infants were assessed for eligibility in the FICare Plus cohort (Figure 2). Of the 64 families approached, 24 parent–infant dyads (37.5%) were enrolled in the FICare Plus group between the two sites. Both cohorts were recruited during COVID-19 pandemic. The NICU policy with respect to parental presence at both hospitals remained the same throughout both phases of the study.

Baseline parent characteristics were similar in both groups (Table 1) except caesarean delivery, which was more common in FICare Plus group (75% vs. 35.2%; *p*-value 0.01). For infants, the median age at enrollment was 8 days which was similar in both groups (8 (5,19) vs. 8 (5.5,13); *p*-value 0.8). No significant difference was observed in gestational age, birth weight, major morbidity and clinical course in infants between the two groups (Table 2).

All FICare Plus families received a link to the on-line educational resources and the FICare Plus parent education booklet. The online educational module was accessed 65 times by 24 families who were enrolled in FICare Plus. Sixteen of the twenty-four families (66%) completed the parent feedback survey. All sixteen parents indicated that they had accessed the on-line educational materials and they agreed or strongly agreed with the statement that the educational materials provided the right kind and right amount of information. Twelve of the sixteen parents responded that they found the information booklet useful.

Of the 25 families who consented to participate in the FICare cohort, 24 families continued their participation throughout their hospital stay. Sixteen FICare Plus families completed the discharge surveys. All FICare Plus families indicated that they enjoyed taking part in the study. The results of the parent feedback survey are presented in Table 3. Despite the COVID-19-related hospital-imposed restrictions, parents in both cohorts indicated that they were able to spend more than 4 h/day with their baby. Of note, all parents in the FICare Plus cohort indicated that they felt ready to interact with their baby when invited to do so and felt confident being involved in their baby’s care, and 87.5% (14/16) of parents indicated that they felt part of their baby’s care team and were able to participate in development of a care plan.

### 3.1. Feasibility

The staff survey was completed by 177 members of the multidisciplinary team involved at both sites (Site A n = 101/200 (50%), Site B n = 76/200 (38%)), and 70% of the respondents indicated that they found the booster education kit helpful. The survey responses indicated that the teams at both sites found FICare Plus acceptable, and it was generally well adopted as presented in Table 4. Of note, there was a significant difference noted between the responses from staff at both sites regarding how satisfying they found the experience of taking care of families enrolled in FICare Plus. In addition, significant differences were noted in the staff survey responses regarding the barriers experienced in implementing FICare between the two sites. More respondents from Site B indicated that inadequate staffing (70 vs. 47%) and limited space to accommodate families (86 vs. 37%) were barriers to FICare Plus implementation. In addition, staff at both sites reported a different role in their pattern of communication with families; more staff in Site A indicated that the bedside nurse would act as a liaison between the family and the clinical team (60%) than at Site B (45%).

### 3.2. Safety

No untoward events in infants, such as accidental extubation, removal of intravenous lines, displacements of nasogastric/orogastric tubes, dislodgement of umbilical catheters, mishandling of equipment, or parental emotional breakdown was observed during the study in FICare Plus group.

### 3.3. Preliminary Measures of Efficacy

The results of the parental measures of efficacy are presented in Table 3. The measures of efficacy for infants, as indicated by clinical outcomes at discharge (Table 5), were similar in both groups. Specifically, there was no difference in the feeding outcome.

The total anxiety scores among parents were similar at enrollment between the two groups (87(67–94) vs. 70.5 (66–86); *p*-value 0.22) (Table 6). However, the scores prior to discharge were lower (less anxiety) in the FICare Plus group as compared to the SC group (78 (71–90) vs. 63 (52–74.5); *p*-value 0.02). This difference did not persist post-discharge. No difference was observed in parental perception of family-centered care (Table 7) and their Perceived Parenting Self-Efficacy Tool scores between FICare Plus and SC groups (Table 8). Similarly, the mean parent stress scores (Table 9) were similar in both groups at enrollment (3.1 vs. 2.8; *p*-value 0.31) and on day 21 (3.3 vs. 3; *p*-value 0.3).

## 4. Discussion

The results of this prospective cohort study suggest that it is feasible and safe to implement FICare in critically ill infants as defined in this study. The tools and procedures provided to staff and families for implementation of FICare Plus resulted in an increased parental perception of their ability to be a part of their infant’s care team, specifically around feeling ready to interact with their baby and being a participant in developing care plans. Our data also suggest that such engagement may have had the positive impact of decreasing parent anxiety prior to discharge. The specific concerns around the risk of increasing parent stress or increasing the risk of safety concerns for the infant were not identified in this pilot study, recognizing that the sample size was very small.

The reflection of healthcare practitioners on their experience of FICare Plus was generally positive, with a recognition of the importance of the parent role in their infant’s care. Interestingly, in the hospital with more complex surgical patients, healthcare practitioners provided a neutral response to the question about whether they found this model of care more satisfying than SC. This merits further exploration. There may be specific characteristics of the staff who chose to work in a surgical NICU that orients their care differently or it may just reflect less familiarity with this model of care and comfort with the skills it requires.

Implementation of FICare in stable term and preterm infants has shown remarkable success across the world. In a multicenter, randomized controlled trial conducted in 26 tertiary care NICUs, implementation of FICare resulted in improved weight gain in infants, higher rates of exclusive breast feeding at discharge and decreased parent stress and anxiety [5]. In another RCT in 11 tertiary care NICUs in China, Hei et al. reported better neonatal outcomes such as shorter hospitalization, shorter duration of oxygen supplementation, reduction in nosocomial infections and less antibiotic use in FICare group infants [6]. In this context, we extended this model of care to both surgical and non-surgical high-risk neonates.

FICare Plus is an adaptation of FICare which is tailored to address the needs of parents of critically ill infants early in their hospital course. Through the engagement of parents and multidisciplinary staff, we developed a specific FICare Plus toolkit including educational resources for staff and parents. We aimed to promote close physical contact between parents and their ill infants, even when they required specialized care due to the severity of their illness. An important task in this regard was to train and support parents to become familiar with the special needs of their infants, such as invasive and non-invasive respiratory support, ostomy bags, catheters, feeding tubes, etc. Although there was a possibility that the expectation placed on parents to perform these tasks might distress them, it was interesting to note that parents felt comfortable in learning and performing these skills when invited to do so as described in their feedback survey. No difference was noted in parent stress on day 21 (3.3 vs. 3; *p*-value 0.3), though this might be due to the small sample size of our study. A recent case-control study [11], supported these findings. They reported that stress scores were significantly lower in parents in the FICare group as compared to those in the control group and that the more parents were involved in their infant’s care, the lower were their stress scores.

In our study, no adverse events, such as accidental extubation, removal of intravenous catheters, displacement of nasogastric tubes or parent emotional breakdown, were observed in the FICare Plus group. This is consistent with the findings of a recent clinical trial conducted in three NICUs in California in which preterm infants of gestational age < 33 weeks with complex medical needs were enrolled. No parent related adverse events were observed in the FICare group in this study [10].

Our findings are also in keeping with those of other recent studies in this population of infants. Moreno-Sanz and her colleagues carried out a study in an advanced NICU which is a referral center for critically ill infants requiring surgeries, extracorporeal membrane oxygenation (ECMO) and other complex procedures. Of the 91 infants enrolled, 74.7% were preterm, 18.7% had complex surgeries and 6.6% had other morbidities. All families enrolled in this FICare study completed the educational module and training and participated regularly in their infant’s care, except one who withdrew voluntarily. Assessment of parental perception about implementation of FICare by both qualitative and quantitative measures (FICare satisfaction questionnaire) showed that parents felt confident and safe to participate in their infant’s care [22]. Similarly, the report from the study conducted in three large neonatal intensive care units (NICUs) including two quaternary care NICUs in California, demonstrated that FICare could be implemented in infants with complex care needs, including those who required surgery. It also showed that implementation of FICare in critically ill preterm babies resulted in improved weight gain and reduction in nosocomial infection [10].

Another interesting finding of our study is that parent anxiety scores prior to discharge were significantly lower in the FICare Plus group as compared to the control group (63 (52–74.5) vs. 78 (71–90); *p*-value 0.02). This is similar to the finding in a multicenter, multinational, cluster-randomized controlled trial in which parents in the FICare group had lower mean anxiety scores as compared to those in the control cohort (70.8 vs. 74.2; *p*-value 0.004) [5].This could be accredited to the fact that they had the opportunity to directly observe and participate in their infant’s care along with the medical team. This may have given them a feeling of self-efficacy and confidence over their parenting abilities resulting in less anxiety. Reduced parent anxiety is a finding of added importance in our study, especially in the context of critically ill infants. NICU-related stress may challenge parents’ abilities to initiate and maintain positive interaction with their infants resulting in anxiety and decreased parent–infant bonding [23]. There is convincing evidence available in the literature which suggests that anxiety influences parenting behaviors with a cumulative long-term negative impact on their child’s development [24,25,26]. This suggests that FICare Plus has the potential to improve not only the short-term outcomes but also provide long-term benefits to infants and their families.

We acknowledge the limitations of our study. Since this study was conducted during the COVID-19 pandemic, education sessions were provided virtually to parents. The research coordinators were unable to have face-to-face interaction with families. In addition, peer support could not be provided to families due to COVID-19-related hospital-imposed restrictions. As a pilot study, with a small sample size, the statistical power was limited. Furthermore, the small sample size also precluded us from comparing the outcomes between the two sites. Future studies with larger sample size are warranted to investigate the impact of FICare Plus on long-term outcomes. Another limitation of our study is that we only included families who could receive education in English, even if it was not their primary language. These findings may therefore not be generalized to our entire NICU population. As FICare is being adapted in many different cultures and situations, further information may become available.

The implications of this study are manifold, with the potential to impact health outcomes, clinical practice, health policy and cost of care. The parents’ comfort with knowing and interacting with their infants even when critically ill may give them confidence and expertise to take better care of their infants. This could result in improved short- and long-term health outcomes for infants, including reduced length of hospital stay. In addition, they may be able to care better after discharge home, with the potential for decreased post-discharge morbidity and reduced healthcare utilization and re-hospitalization. All these may reduce the cost of NICU care and thus have important implications for both policy development and resource allocation decisions.

## 5. Conclusions

This pilot study suggests that it is feasible and safe to implement family-integrated care in critically ill infants. Future quality improvement initiatives may provide further information about the processes required for its sustainability and refinement.

## Figures and Tables

**Figure 1 children-10-01337-f001:**
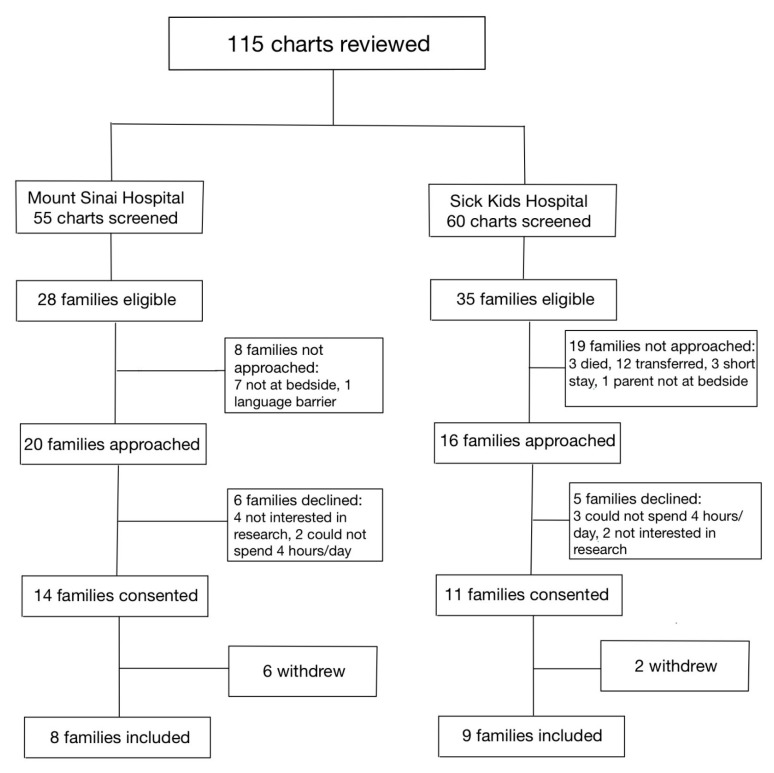
Flow diagram of families included in standard care cohort.

**Figure 2 children-10-01337-f002:**
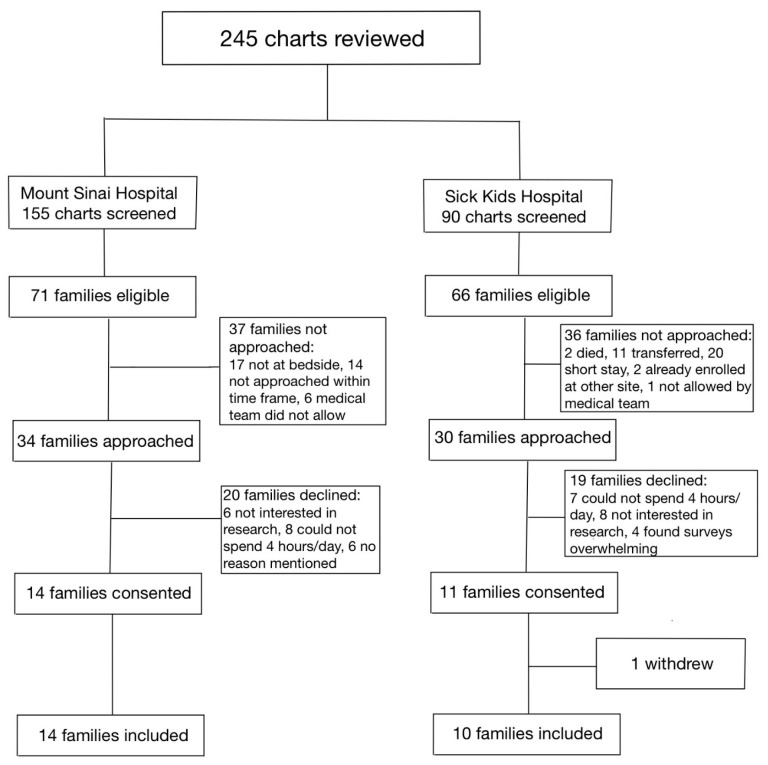
Flow diagram of families included in FICare Plus cohort.

**Table 1 children-10-01337-t001:** Baseline parental characteristics.

	Standard Care (17)	FICare Plus (24)	*p*-Value
Relationship, %(n/N)			0.99
Mother	82.35 (14/17)	83.3 (20/24)	
Father	17.65 (3/17)	16.6 (4/24)	
English as first language, %(n/N)	52.94 (9/17)	70 (14/20)	0.29
Education level (Bachelor or above), %(n/N)	35.29 (6/17)	63.16 (12/19)	0.09
Employment (or self-employment), %(n/N)	82.35 (14/17)	83.3 (20/24)	0.09
Prenatal Care (yes), %(n/N)	52.94 (9/17)	62.5 (15/24)	0.16
Parental age (years), median (IQR)	30 (26, 33)	31.5 (29.5, 36.5)	0.12
Parental self-rated health status (0–100), median (IQR)	97 (77.5, 100)	97 (85.5, 100)	0.77
Singleton conception %(n/N)	70.59 (12/17)	83.33 (20/24)	0.45
Primigravida mothers, %(n/N)	29.41 (5/17)	20.83 (5/24)	0.71
Cesarean delivery, %(n/N)	35.29 (6/17)	75 (18/24)	0.01
Abortions, %(n/N)	41.18 (7/17)	58.33 (14/24)	0.27
Out born delivery, %(n/N)	64.71 (11/17)	41.67 (10/24)	0.14

**Table 2 children-10-01337-t002:** Baseline infant characteristics and their clinical parameters.

	Standard Care	FICare Plus	*p*-Value
Number of infants	17	24	
Gestational age (weeks), median (IQR)	26.05 (24.2, 29)	28.9 (25.5, 32.1)	0.15
Age at enrolment (days), median (IQR)	8 (5, 19)	8 (5.5–13)	0.8
Birth weight (kg), median (IQR)	0.71 (0.62, 0.82)	0.93 (0.74, 1.63)	0.07
Sex, %(n/N)	52.94 (9/17)	50 (12/24)	0.85
Apgar score < 7 at 1 min, %(n/N)	71.43 (10/14)	50 (12/24)	0.19
SNAPII score > 20, %(n/N)	35.29 (6/17)	20.83 (5/24)	0.47
Intestinal Perforation, %(n/N)	11.76 (2/17)	8.33 (2/24)	0.99
Surgical intervention, %(n/N)	35.29 (6/17)	37.5 (9/24)	0.89
Sepsis, %(n/N)	35.29 (6/17)	16.67 (4/24)	0.27
Esophageal atresia	17.65 (3/17)	29.17 (7/24)	0.47
Invasive ventilation days, median (IQR)	6 (1, 44)	3 (0, 6)	0.09
NIPPV ventilation days, median (IQR)	4 (0, 14)	3 (0, 18.5)	0.96
CPAP days, median (IQR)	15 (6, 37)	21 (12, 30)	0.75

**Table 3 children-10-01337-t003:** Parent feedback survey.

	Standard Care	FICare Plus
Able to spend > 4 h/day with my baby % n/N	90.91 (10/11)	93.75 (15/16)
Feel confident being involved in my baby’s care % n/N	90.91 (10/11)	100 (16/16)
Feel like I know my baby % n/N	100 (11/11)	100 (16/16)
Felt ready to interact with my baby when invited to do so % n/N	63.64 (7/11)	100 (16/16)
Found participating in rounds helpful % n/N	90.91 (10/11)	100 (16/16)
Felt part of my baby’s care team % n/N	81.82 (9/11)	87.5 (14/16)
Was able to participate in development of care plan % n/N	63.64 (7/11)	87.5 (14/16)
Had enough information on baby’s plan of care at discharge % n/N	80 (8/10)	85.71 (12/14)

**Table 4 children-10-01337-t004:** Staff feedback survey.

	Site A	Site B	*p*-Value
I became more of a mentor to the parents rather than a direct care provider.			0.4
Very/Somewhat true % n/N	56.44 (57/101)	56.58 (43/76)	
Neutral % n/N	32.67 (33/101)	26.32 (20/76)	
Somewhat untrue/Not at all true % n/N	10.89 (11/101)	17.11 (13/76)	
I felt I was taking care of the entire family not just the baby.			0.88
Very/Somewhat true % n/N	83.17 (84/101)	85.9 (67/78)	
Neutral % n/N	10.89 (11/101)	8.97 (7/78)	
Somewhat untrue/Not at all true % n/N	5.94 (6/101)	5.13 (4/78)	
I found taking care of families in the FICare Plus model more challenging than taking care of SC families.			0.86
Very/Somewhat true % n/N	32.67 (33/101)	28.95 (22/76)	
Neutral % n/N	31.68 (32/101)	34.21 (26/76)	
Somewhat untrue/Not at all true % n/N	35.64 (36/101)	36.84 (28/76)	
I found taking care of families in the FICare Plus model more satisfying.			0.028
Very/Somewhat true % n/N	69.31 (70/101)	53.25 (41/77)	
Neutral % n/N	21.78 (22/101)	40.26 (31/77)	
Somewhat untrue/Not at all true % n/N	8.91 (9/101)	6.49 (5/77)	
I have seen a positive effect on the parent-infant relationship.			0.95
Very/Somewhat true % n/N	81.19 (82/101)	81.25 (65/80)	
Neutral % n/N	15.84 (16/101)	15 (12/80)	
Somewhat untrue/Not at all true % n/N	2.97 (3/101)	3.75 (3/80)	
I have seen a positive effect of this model on the parent healthcare team relationship.			0.55
Very/Somewhat true % n/N	78 (78/100)	70.89 (56/79)	
Neutral % n/N	18 (18/100)	24.05 (19/79)	
Somewhat untrue/Not at all true % n/N	4 (4/100)	5.06 (4/79)	
I feel overall that this model of care allows for greater scope on a nurse’s role in the NICU.			0.08
Very/Somewhat true % n/N	66.34 (67/101)	61.54 (48/78)	
Neutral % n/N	22.77 (23/101)	34.62 (27/78)	
Somewhat untrue/Not at all true % n/N	10.89 (11/101)	3.85 (3/78)	

**Table 5 children-10-01337-t005:** Clinical outcomes at discharge.

	Standard Care	FICare Plus	*p*-Value
Number of infants	17	24	
Gavage, % (n/N)	60 (9/15)	70.83 (17/23)	0.49
Breast milk only, % (n/N)	53.33 (8/15)	60.87 (14/23)	0.64
Formula milk only, % (n/N)	6.67 (1/15)	8.70 (2/23)	0.99
Both Breast and Formula and milk, % (n/N)	33.33 (5/15)	30.43 (7/23)	0.99
Weight at discharge (kg), median (IQR)	3.24 (2.03, 3.65)	2.64 (1.71, 2.92)	0.22
Discharge destination, % (n/N)			0.26
Home	23.53 (4/17)	12.5 (3/24)	
Inpatient area	29.41 (5/17)	16.67 (4/24)	
Other hospitals (NICU or Community)	35.29 (6/17)	66.67 (16/24)	
Died before discharge	11.76 (2/17)	4.17 (1/24)	

**Table 6 children-10-01337-t006:** Total anxiety scores.

	Standard Care	FICare Plus	*p*-Value
At enrollment, median (IQR)	87 (67, 94)	70.5 (66, 86)	0.22
At Day 21, median (IQR)	88.5 (71, 103)	78.5 (63, 91)	0.3
Prior to Discharge, median (IQR)	78 (71, 90)	63 (52, 74.5)	0.029
Post Discharge, median (IQR)	73 (65, 88)	70.5 (49, 77)	0.35

**Table 7 children-10-01337-t007:** Perception of family-centered care.

	Standard Care	FICare Plus	*p*-Value
At enrollment, median (IQR)	5.56 (4.89, 6.56)	6.22(5.78, 7)	0.12
Prior to Discharge, median (IQR)	5.11 (3.89, 5.89)	5.33 (4.89, 5.78)	0.68

**Table 8 children-10-01337-t008:** Perceived parenting self-efficacy scores.

	Standard Care	FICare Plus	*p*-Value
At day 21, mean (SD)	58.13 (10.86)	62.44 (6.33)	0.18
Prior to Discharge, mean (SD)	65.7 (8.04)	65.69 (7.07)	0.99
Post discharge, mean (SD)	66.44 (7.84), 9	66.67 (6.41), 8	0.94

**Table 9 children-10-01337-t009:** Parent stress scores.

	Standard Care	FICare Plus	*p*-Value
At enrollment, mean (SD)	3.10 (0.96)	2.81 (0.74)	0.31
At day 21, mean (SD)	3.32 (0.93)	3.0 (0.62)	0.30

## Data Availability

The data presented in this study are not publicly available due to privacy restrictions.

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
