# Peer review of "A Pilot Study of Family-Integrated Care (FICare) in Critically Ill Preterm and Term Infants in the NICU: FICare Plus"

_children, 2023, doi:10.3390/children10081337_

Round 1

Reviewer 1 Report

Manuscript deals with the description of FICare, a intervention aimed to improve the state of infant and parent during hospitalization. given the clinical implication, the topic is relevant.

Nevertheless, structure of manuscript had several limitation, that impair the quality of the work.

Abstract is not focused. in first sentences the new approach (FICare plus) should be declared, or reader will think about other approach. some info should be given precisely (i.e name of questionnaires). specific results are missing on benefit/avalilability of intervention. conversely, some info (i.e descriptive characteristics) could be postponed to main text.

Introduction should be improved. introduction section should clearly drive reader to the knowldege of the state of art, and what is missing, in order to show the relevance of the aim. The description of the studies investigated the benefit of FICare during infant hospitalization, and then the need to implement this program also for severe infants is somehow confousing, and it is not clear which the population already investigated and which are not. Furthermore, I think that an explanation of the specificity of FICare compare to other parent program (i.e. NIDCAP) should be given.

moreover, authors will consider many effects on infants, parent (anxiety, distress, sense of competence, ...) and staff. please describe the state of art on all these variables.

the main concern in this section is the absence of aim, that is declared in the method section. 

Method. Group selection is problematic. First, groups are different for size, time of assessment (12 vs 6 months) and, especially, are recruited in peculiar moment. SC is recruited during the first wave of covid-19, FICare Plus group was recruited in also critical period but when population were adapting to restriction (or it was in my country). Authors should specify if hospital policy differed in these periods.

moreover, the two sites are different for patients severity. authors should consider to compare if samples from these two sites differ into results.

Procedure describe the work done for the improvement of FIcare plus. I suggest to better explain the differences  form standard FICare. moreover, it is not clearly explain which instruments were admistered and why.

I also suggest a section specific for instruments.

for analyses section, I think that for variables that have many administration (i.e anxiety, distress, etc), a repeated measure is a more adequate analyses.

results should be improved, given a sub-division in sub section, according to different aim. I also suggest a better improve of formatting of tables.

Author Response

Thank you very much for reviewing our manuscript and for the valued comments. We appreciate the quality of these comments and have made changes in the manuscript based on the suggestions. Please find below a point-by-point response to the comments:

  1. Abstract is not focused. in first sentences the new approach (FICare plus) should be declared, or reader will think about other approach. some info should be given precisely (i.e name of questionnaires). specific results are missing on benefit/avalilability of intervention. conversely, some info (i.e descriptive characteristics) could be postponed to main text.

Response: We thank the reviewer and agree with the above comment. To address this, we have revised the abstract and have added the following:

In line 13-21: “an augmented FICare program termed FICare Plus, in critically ill infants in the first few weeks of life. Resources were specifically developed for staff and parents to support earlier parental engagement in infant care. The primary aim was to assess if it is feasible to support greater parental engagement in taking care of critically ill infants, given the tools and procedures developed for FICare Plus, as determined by responses on parent and staff surveys. Infant health outcomes and standardized measures of parental stress, anxiety and parenting self-efficacy were also collected. The t-test or Wilcoxon Rank-Sum test were used to compare continuous variables, while Chi-square or Fisher exact test were used for categorical variables, respectively. “

In line 24-25: “Parents in FICare Plus cohort felt confident to participate in their infants’ care. The staff also found this model of care acceptable and well adopted.”

The standardized questionnaires used were State -Trait Anxiety Inventory (STAI), Parental Stressor Scale: NICU (PSS: NICU), Perceived Parenting Self- Efficacy Tool and Family Centered Care Survey (FCC). Adding the names of these questionnaires would exceed the word count for abstract so they are mentioned in the text instead of abstract.

  1. 2. Introduction should be improved. introduction section should clearly drive reader to the knowldege of the state of art, and what is missing, in order to show the relevance of the aim. The description of the studies investigated the benefit of FICare during infant hospitalization, and then the need to implement this program also for severe infants is somehow confousing, and it is not clear which the population already investigated and which are not. Furthermore, I think that an explanation of the specificity of FICare compare to other parent program (i.e. NIDCAP) should be given. moreover, authors will consider many effects on infants, parent (anxiety, distress, sense of competence, ...) and staff. please describe the state of art on all these variables. the main concern in this section is the absence of aim, that is declared in the method section. 

Response: Thank you for this comment. The current state of knowledge about the benefits of FICare in different patient populations is mentioned in our manuscript in lines 42-45.

We have revised the introduction section according to your guidance.

a) A statement has been added to clarify how FICare is different from other parent programs such as NIDCAPin lines 36-40:

“FICare is a care delivery model that addresses the structure and processes of care delivery in Neonatal Intensive Care Units (NICU). It provides a framework for parent-partnered care delivery that can include multiple specific parent-partnered interventions focused on different aspects of care, for example feeding and infant development [2]”

b)We have added the following text indicating what is missing in our current knowledge in lines 67-77:

“Although it may have seemed like a natural extension to provide FICare to all families, concerns were brought forward by both staff and parent advisors as to whether this was the correct approach for parents of critically ill infants. Parents brought forward concerns that the expectations around early engagement in their infants’ care might be overwhelming for them. Staff were concerned about increasing parental anxiety and the possibility of increased medical risk to the infants of parent engagement while still unstable. There were no published data in this population at the time of study conception. Facilitating infant-parent interaction creates a consistent care environment which is beneficial for both infants and their parents [5]. Moreover, parental confidence and expertise in taking care of their infants, particularly if they have complex needs, may lead to better long-term neurodevelopmental outcomes in their infants [8].”

c)To clearly state the aim of the study in introduction part, we have added the following in lines 79-84:

“Therefore, we hypothesized that, despite these concerns, parents of critically ill infants might in fact be the group who, if well supported, may benefit more from FICare. This study was conducted with the objective to assess if it is safe and feasible to support greater parent engagement in taking care of their critically ill infants, without increased parental stress and anxiety or increasing risk of harm to infants.”

3.Method. Group selection is problematic. First, groups are different for size, time of assessment (12 vs 6 months) and, especially, are recruited in peculiar moment. SC is recruited during the first wave of covid-19, FICare Plus group was recruited in also critical period but when population were adapting to restriction (or it was in my country). Authors should specify if hospital policy differed in these periods.

moreover, the two sites are different for patients severity. authors should consider to compare if samples from these two sites differ into results.

Response: We recognize that the two groups are different in terms of number of infants, but the baseline infant characteristics were similar in both groups as mentioned in Table 2. Following statement has been added in lines 219-221:

“Both cohorts were recruited during COVID pandemic. The NICU policy with respect to parental presence at both hospitals remained same throughout both phases of the study.”

Since the two sites have different patient population and different level of experience with FICare, we observed a significant difference in staff survey response at two sites which is already mentioned in manuscript on lines 277-284 as:

“In addition, significant differences were noted in the staff survey responses regarding the barriers experienced in implementing FICare between the two sites. More respondents from Site B indicated that inadequate staffing (70 vs 47%) and limited space to accommodate families (86 vs 37%) were barriers to FICare Plus implementation. In addition, staff at both sites reported a different role in their pattern of communication with families; more staff in site A indicated that the bedside nurse would act as a liaison between the family and the clinical team (60%) than at Site B (45%).”

The possible reasons which could potentially contribute to this difference in results are mentioned in the discussion section in lines 331-334.

The sample size of enrolled infants was too small to draw any conclusions about the infant outcomes between the two groups.

  1. Procedure describe the work done for the improvement of FIcare plus. I suggest to better explain the differences form standard FICare. moreover, it is not clearly explain which instruments were admistered and why

Response: Thanks for this suggestion. Following sentences have been added in lines 146-148:

“The SC cohort received the usual care at both sites.  FICare Plus training was provided to nursing staff after the accrual of SC cohort, so as not to influence the SC provided to this group.”

As for a section about FICare Plus resources /instruments (as mentioned in the comment), we have added a subheading “FICare Plus educational toolkit for staff and parents” in line 128. The subsequent paragraph under this subheading provides details about the resources/instruments which were developed for staff and parents for implementation of FICare Plus.

  1. For analyses section, I think that for variables that have many administration (i.e anxiety, distress, etc), a repeated measure is a more adequate analysis.

Response: Thank you for this comment. We have discussed this point with our statistician. As our secondary research question was to investigate the effect of  FiCare (between pre vs post implementation) and not the prolonged effects of FiCare, we consider pre vs post comparison sufficient for this purpose.

6.results should be improved, given a sub-division in sub section, according to different aim. I also suggest a better improve of formatting of tables.

Response: We thank the reviewer for this comment and agree with the suggestion. In results section, subheadings based on the aims of the study are mentioned as “Safety” in line 287, “Preliminary measures of efficacy” in line 292. We have now added another subheading as “Feasibility” in line 270.

We have reformatted all the tables.

Reviewer 2 Report

The authors have aimed to assess the safety and feasibility  of implementation of family integrated care (FIC) in critically ill infants.  The authors conclude that it was feasible and safe to implement FIC.

Comments:

1. The authors do not give a sample size calculation to determine if they could detect important differences in the outcomes - given the heterogeneity in the  populations of the two centres they report, this is particularly  important. 

2. This study was undertaken during COVID and hence whether the results are generalisable needs to be questioned. 

3. There is loss to complete the study, it would be important to see if those lost were comparable to those who completed the study.

4. The authors, given the limitations of the study they acknowledge, over state the implications of their results. 

Minor comments:

1. "Data" is a plural word. 

Author Response

Thank you very much for reviewing our manuscript and for the valued comments. We appreciate the quality of these comments and have made changes in our manuscript based on the suggestions. Please find below our point-by-point response to the comments:

  1. The authors do not give a sample size calculation to determine if they could detect important differences in the outcomes - given the heterogeneity in the  populations of the two centres they report, this is particularly  important. 

Response: We thank the reviewer for this comment. This was a pilot study with a small sample. The purpose of a pilot study is not to detect a meaningful difference (although this could be hinted from the analysis in a pilot study), but rather, to see the feasibility for a full study. Detection of difference with meaningful effect size can be investigated in a full study. Sample size calculation is needed for a full study, and the assumptions needed for the sample size calculation can be obtained from a pilot study. Therefore, one of the reasons for conducting a pilot study is to get the assumptions required for calculation of sample size for a full study. Therefore, a pilot study normally does not require sample size calculation (especially when we don't know the assumptions which are used for the sample size calculation), it just needs to reasonably represent the target population.

  1. This study was undertaken during COVID and hence whether the results are generalisable needs to be questioned.

Response: Thanks for this comment. We recognize this limitation. However, despite the COVID restrictions, parents complied with the expectation required for implementation of FICare Plus. This shows that parents are keen to get involved in their infants’ care and despite many hurdles during COVID lockdown period, they managed to spend time with their infants and get involved in their infants’ care. Since FICare was well-accepted under restricted conditions, we think it will be more easily accepted and implemented under normal circumstances and so the results can be generalised to some extent. However, we acknowledge that this study was conducted during COVID pandemic and have added the following statement as a limitation of our study in lines 397-401:

“Since this study was conducted during COVID pandemic, education sessions were provided virtually to parents. The research coordinators were unable to have face-to-face interaction with families. In addition, peer support could not be provided to families due to COVID-19 related hospital-imposed restrictions”.

  1. There is loss to complete the study, it would be important to see if those lost were comparable to those who completed the study.

Response: All 24 families who were recruited in FICare Plus group continued their participation throughout their hospital stay. However, discharge surveys were completed by sixteen families. The discharge surveys were filled online anonymously by the participants. Hence, it is not known which families completed and which did not complete the survey and so the comparison cannot be done.

  1. The authors, given the limitations of the study they acknowledge, over state the implications of their results. 

Response: We agree with this comment. In our conclusion section, we have deleted the word “show” and replaced it with the word “suggest” in line 416.

5.Minor comments: "Data" is a plural word. 

Response: We apologize for this oversight. Following changes have been made in the manuscript:

In line 99 and 204, the word “was” replaced by the word “were”.

In line 323, the word “suggests” replaced by the word “suggest”.

Reviewer 3 Report

Thank you for submitting your manuscript entitled “A pilot study of Family Integrated Care (FICare) in Critically ill preterm and term infants in the NICU: FICare Plus”. I have carefully reviewed the manuscript. I want to provide you with feedback and recommendations for improvement.

Please find below my comments:

·      Line 15: Replace “(SC)and” with “(SC) and”.

·      Line 49: The authors have to use the term “late preterm” instead of “near term” and add a definition and a reference.

·      Line 59-61: Please address the potential FICare benefits for critically ill infants. Explain why it is important to develop a FICare for critically ill neonates.

·      Line 66: Define the meaning of “infants of all gestational age”. If the authors mean preterm, late-term, or full-term, they should present the results in different neonatal-age groups. They cannot compare a critically ill neonate of 28 weeks to a critically ill neonate of 40 weeks.

·      Line 68-74: If the authors make a comparison, they should refer to Site A and Site B by specific variants. For example, they cannot present the percentage of low-body-weight infants in Site A and then focus on infants that underwent surgery on Site B.

·      Line 126: The authors referred to “Between Jan 1 to July 1, 2021,” while the results and Line 187 referred to “ Between Jan 1 to July 31, 2021”, a different time frame. They have to correct it.

·      Line 134-146: This situation should also be referred to as a study limitation because lack of face-to-face interaction can lead to different outcomes regarding behavioral experience and performance of neonatal care.

·      Line 161-165: Replace “Parent measures” and “parent stress” with “Parental measures”, “parental stress” etc.

·      Line 162: Define the difference between stress and anxiety and their effects on neonate care.

·      Line 256: Replace “parent measures” with “parental measures”.

·      Line 283: Replace the word “suggests” with “suggest”.

·      Line 302:  Present the effect on neonates’ health.

·      Line 316: Replace “et al” with “et al.,”.

·      Line 341-354: Correlate parental stress with the neonates' health outcome.

·      Line 353: Replace “short–term” with “short-term”.

As a general comment, you can focus on the effect of FICare on severely ill neonates' health outcomes so you present better the innovation and the benefit of this study.

These suggestions will help you improve your manuscript. 

Minor editing of the English language required

Author Response

Thank you very much for reviewing our manuscript and for the valued comments. We appreciate the quality of these comments and have made changes in our manuscript based on the suggestions. Please find below our point-by-point response to the comments: 

Line 15: Replace “(SC)and” with “(SC) and”.

Response: We apologize for this oversight. We have corrected it in our manuscript in line 22 in the revised manuscript.

Line 49: The authors have to use the term “late preterm” instead of “near term” and add a definition and a reference.

Response:  Thanks for this comment. We acknowledge this mistake.

We have described the findings of a study which included term and preterm infants of gestational age 26 weeks and above, we have replaced the word “near-term” by the word “preterm” in line 58 of the revised manuscript.

Line 59-61: Please address the potential FICare benefits for critically ill infants. Explain why it is important to develop a FICare for critically ill neonates.

Response: We have added the following statements in lines 74-77 in the revised manuscript to highlight the potential benefits of FICare plus for critically ill infants.

“Facilitating infant-parent interaction creates a consistent care environment which is beneficial for both infants and their parents [5]. Moreover, parental confidence and expertise in taking care of their infants, particularly if they have complex needs, may lead to better long-term neurodevelopmental outcomes in their infants [8].”

Line 66: Define the meaning of “infants of all gestational age”. If the authors mean preterm, late-term, or full-term, they should present the results in different neonatal-age groups. They cannot compare a critically ill neonate of 28 weeks to a critically ill neonate of 40 weeks.

Response: Thanks for highlighting this point.  The focus of our research study was mainly on the parents and the NICU staff. The aim was to determine if, with the resources provided to them, it was feasible to implement FICare in critically ill infants. Therefore, we did not compare infant outcomes but rather the experience of their parents and nursing staff.

Line 68-74: If the authors make a comparison, they should refer to Site A and Site B by specific variants. For example, they cannot present the percentage of low-body-weight infants in Site A and then focus on infants that underwent surgery on Site B.

Thanks for this comment.

“Infants eligible for this study included: ELBW infants on invasive positive pressure ventilation for >48 hours after birth, infants with surgical necrotizing enterocolitis or bowel perforation, infants with tracheoesophageal fistula or esophageal atresia”.

As mentioned in the previous response, we did not compare infants at both sites, we rather compared the experience of the families and staff at both sites.

Line 126: The authors referred to “Between Jan 1 to July 1, 2021,” while the results and Line 187 referred to “ Between Jan 1 to July 31, 2021”, a different time frame. They have to correct it.

Response: We apologize for this mistake. In line 152 of the revised manuscript, we have corrected it to “Between Jan 1 to July 31, 2021”.

Line 134-146: This situation should also be referred to as a study limitation because lack of face-to-face interaction can lead to different outcomes regarding behavioral experience and performance of neonatal care.

Response: We agree with this suggestion. We have added the following statements in lines 397-401 of the revised manuscript:

“Since this study was conducted during COVID pandemic, education sessions were provided virtually to parents. The research coordinators were unable to have face-to-face interaction with families. In addition, peer support could not be provided to families due to COVID-19 related hospital-imposed restrictions”.

Line 161-165: Replace “Parent measures” and “parent stress” with “Parental measures”, “parental stress” etc.

Response: We have replaced the above terms in our manuscript.

Line 162: Define the difference between stress and anxiety and their effects on neonate care.

Response: Thank you for this suggestion. We have added the following statement with reference in line 190-193 of the revised manuscript to define stress and anxiety:

“Stress is defined as a physiological response to a stimulus which is accompanied by a simultaneous emotional reaction whereas anxiety is defined as “the anticipation of a future threat” by The Diagnostic and Statistical Manual of Mental Disorders, Fifth Edition (DSM-5) [17]”

The effects of parental stress and anxiety on their infants is described in the manuscript in discussion section in lines 392-394.

Line 256: Replace “parent measures” with “parental measures”.

Response: We apologize for this mistake. It has been corrected in line 293 of the revised manuscript.

Line 283: Replace the word “suggests” with “suggest”.

Response: We have replaced it in line 323 of the revised manuscript

 Line 302:  Present the effect on neonates’ health.

The sample size of enrolled infants in our study was too small to draw any conclusions about the infant outcomes.

Line 316: Replace “et al” with “et al.,”.

Response: Replaced in the manuscript  in line 356.

Line 341-354: Correlate parental stress with the neonates' health outcome.

Response: Thanks for this comment. We have added the following statement in line 390-392 of the revised manuscript:

“NICU-related stress may challenge parents’ abilities to initiate and maintain positive interaction with their infants resulting in anxiety and decreased parent-infant bonding [23].”

This is followed by the statement in manuscript in lines 392-394.        

“There is convincing evidence available in literature which suggests that parental anxiety influences parenting behaviors with a cumulative long-term negative impact on their child’s development [24-26]”.

Line 353: Replace “short–term” with “short-term”.

Response: We have replaced it in the manuscript.

As a general comment, you can focus on the effect of FICare on severely ill neonates' health outcomes so you present better the innovation and the benefit of this study.

Response: We thank you for your valued comment. The aim of our study was to assess the safety and feasibility of FICare Plus and not to assess its effects on neonatal outcomes. The sample size of our study was not large enough to comment on the impact of FICare Plus on the infants’ health outcomes.

Round 2

Reviewer 1 Report

I thank authors for their work that improved the quality of manuscript.

I had only some few obeservation. 

I understand that abstract has a limit of word, and that it is hard to include all the info of a manuscript. However, The info on methodology are mandatory and are needed to a better understanding of the work. Moreover, they could lead reader to read (or buy) an article and to cite it. So, I suggest again to include the name of the instrument. I suggest to reduce the long sentence of background, that also made somehow disomogenous the abstract (only few line for method, result and conclusion).  

Again, I well-know the difficulties in recruit wide sample in nicu patients (fortunely!) and that limited the inclusion of comparison between sites, or the use of repeated measures. It is ok that authors chose to did not analyse this. However, I suggest to better specify this difficulties in limit section, considering them as potential further investigation.

finally, in the manuscript downloaded, only some tables (3 and 4) are formatted, but the other ones were similar to previous version. I suggest to make them homogeneous.

Author Response

We thank the reviewer for reviewing our manuscript again and for the valued suggestions and comments. Please find below our point-by-point response  to the comments:

I understand that abstract has a limit of word, and that it is hard to include all the info of a manuscript. However, The info on methodology are mandatory and are needed to a better understanding of the work. Moreover, they could lead reader to read (or buy) an article and to cite it. So, I suggest again to include the name of the instrument. I suggest to reduce the long sentence of background, that also made somehow disomogenous the abstract (only few line for method, result and conclusion).  

Response: We thank the reviewer for this comment and agree with the suggestion. To include the names of questionnaires/instruments used, we have made the following changes in the manuscript:

  1. The statement below is deleted in lines 15-18:

“The primary aim was to assess if it is feasible to support greater parental engagement in taking care of critically ill infants, given the tools and procedures developed for FICare Plus, as determined by responses on parent and staff surveys.

  1. Following statement is added in lines 17-19 of the revised manuscript to include the names of instruments/questionnaires:

“using standardized questionnaires: State -Trait Anxiety Inventory (STAI), Parental Stressor Scale: NICU (PSS: NICU), Perceived Parenting Self- Efficacy Tool and Family Centered Care Survey (FCC).”

Again, I well-know the difficulties in recruit wide sample in nicu patients (fortunely!) and that limited the inclusion of comparison between sites, or the use of repeated measures. It is ok that authors chose to did not analyse this. However, I suggest to better specify this difficulties in limit section, considering them as potential further investigation.

Response: Thanks for this comment. We have added the following statement as another limitation of our study in lines 401-403 of the revised manuscript:

“Furthermore, the small sample size also precluded us from comparing the outcomes between the two sites. Future studies with larger sample size are warranted to investigate the impact of FICare Plus on long-term outcomes.”

finally, in the manuscript downloaded, only some tables (3 and 4) are formatted, but the other ones were similar to previous version. I suggest to make them homogeneous.

Response: Thanks for this suggestion. We have formatted the tables to look homogenous. However, if they look different, we request editorial support to format them as per journal’s requirements.

Reviewer 2 Report

   The authors have addressed my comments.  

Author Response

We thank the reviewer for accepting our comments.